# Menstrual hygiene management and menstrual secrecy among young women in rural Lao PDR: A cross-sectional study

Kanayo Ono[1], Yu Sato[2,3]*, Noriko Kuwano[1], Hisao Ando[2,3], Kana Maruyama[1]

**1** Graduate School of Oita University of Nursing and Health Sciences, Oita, Japan, **2** St. Mary's Hospital, Our Lady of the Snow Social Medical Corporation, Kurume, Japan, **3** Non-Profit Organization ISAPH, Tokyo, Japan

☯ These authors contributed equally to this work.

* yunyunyun@live.jp

**Data Availability Statement:** All relevant data are within the manuscript and its Supporting Information files.

## Abstract

As a fundamental right, all women should have equal access to menstrual hygiene management (MHM). However, certain sociocultural contexts foster an atmosphere of secrecy surrounding menstruation, which discourages open discussion. The present study seeks to explore the relationship between attitudes on menstrual secrecy and MHM practices among young women in rural areas of the Lao People's Democratic Republic (Lao PDR). In March 2023, a cross-sectional survey using semi-structured self-administered questionnaires was conducted on 80 women (age range, 15–24 years) randomly selected from eight villages in the central part of Lao PDR. Logistic regression analysis was performed to examine the associations between sociodemographic and economic characteristics, MHM practices, social support, and attitudes toward secrecy regarding menstruation. Among the 70 respondents, 68 (97.1%) reported being satisfied with current MHM practices; however, 27 (38.6%) agreed that menstruation should not be discussed with others. Some women lacked access to private spaces for changing pads or washing. The results of the logistic regression analysis indicated that women with higher monthly disposable income were less likely to endorse menstruation secrecy (odds ratio: 0.15, 95% confidence interval: 0.02–0.85). No significant associations were found between MHM practices and attitudes on secrecy. Although no direct association was found between MHM practices and attitudes on menstruation secrecy, some participants remain in unfavorable MHM environments.

## Introduction

Menstruation is a universal physiological phenomenon common to humanity throughout the reproductive period, excluding pregnancy, postpartum, and lactation, from menarche to menopause, and is a healthy part of everyday life [1, 2]. Safely and hygienically managing the monthly menstrual cycle with dignity while protecting privacy is a fundamental right for all women [3]. It is currently estimated that 1.9 billion people worldwide experience menstruation, but about 500 million of these individuals are unable to achieve adequate menstrual

**Funding:** The author(s) received no specific funding for this work.

hygiene management (MHM) [4]. Reports on menstrual poverty are widespread worldwide, including restrictions on access to menstrual products, menstrual education, and proper water and sanitation facilities [5, 6]. Furthermore, issues related to MHM extend beyond health and hygiene consequences alone. High school dropout rates, associated with gender equality [7], social stigma surrounding menstruation, education, social participation, and potential economic opportunities being lost, have been reported among girls, making the attainment of proper MHM for all women an urgent priority [4].

Issues regarding MHM are concentrated in low- and middle-income countries (LMICs), and one of the underlying factors is a lack of knowledge and misunderstanding about menstruation [8]. Among the reasons for this is the underdevelopment of health and hygiene education, including education related to menstruation. A recent review of 44 studies concluded that adolescent girls generally have insufficient knowledge about menstruation [9]. Because of insufficient information, guidance, and support, many girls living in Africa, Asia, and Latin America fear menstruation excessively and do not seek necessary assistance [10]. Regarding the implementation of proper MHM, there have also been reports of an association with economic factors. With Asia's population estimated at 4.4 billion, regions such as South Asia and Southeast Asia have attracted significant attention because of their remarkable economic growth rates; however, more than 500 million people still live below the poverty line. In households facing economic issues, access to menstrual products may be restricted, and obtaining materials to maintain hygiene can become difficult [11].

Furthermore, sociocultural and religious factors can be determinants of MHM practice. Perceptions of menstruation as impure or shameful have been confirmed in various regions worldwide, and girls leant sociocultural attitude that they barred from participating in social and religious activities [12]. On the other hands, discussing menstruation is rare in some areas, even among women, with only 26.2% of young women reported to have received information about menstruation from their mothers [13]. Moreover, because these values rely on sociocultural contexts, it is noted that disseminating knowledge alone may be insufficient for behavioral change [13]. Therefore, to achieve proper MHM for all women, it is necessary to understand the complex interplay of political, social, cultural, and economic influences and to take active measures to address them.

The Lao People's Democratic Republic (Lao PDR) is one of the poorest LMICs in Southeast Asia. Despite being classified as a middle-income country based on its gross national income, social development lags behind, suggesting its status as a least-developed country. According to government surveys, disparities in MHM based on various "gaps" have been reported [14]. For example, among women experiencing menstruation, the proportion of those with access to adequate menstrual products and home washrooms/dressing rooms differs between rural and urban areas (88.5% vs. 45.5%, respectively), women with no education versus those with secondary education (35.1% vs. 89.8%, respectively), and low- versus high-income households (20% vs. 94.2%, respectively). Furthermore, previous studies have reported that adolescent girls face limitations in social participation/school attendance because of menstruation [15, 16]. The implementation of proper MHM is not only essential for protecting women's rights, but also beneficial for supporting women's participation in society.

In Lao PDR, there is a known cultural perception of menstruation as impure, and in some regions, the use of menstrual huts during menstruation has been reported [3]. Additionally, it has been suggested that adolescent girls may avoid talking about menstruation to avoid period shaming by males [17]. Given these social and cultural backgrounds, it is conceivable that a cultural value of secrecy surrounding menstruation exists in Lao PDR, which, in turn, raises concerns about access to support for menstruation and limitations on behavior based on

correct information. In other words, the sociocultural context surrounding menstruation may hinder the implementation of proper MHM.

Given this background, the present study focuses on menstruation secrecy and the realities of MHM among rural adolescent girls in Lao PDR, for whom sufficient reporting is lacking. The aim of this study is to investigate how attitudes of secrecy toward menstruation are related to the elements constituting MHM. The results may provide insights into which aspects need to be considered for promoting MHM in addition to information for future initiatives.

## Materials and methods

### Study design

The present cross-sectional study utilized anonymous self-administered semi-structured questionnaires.

### Study site and target population

The study area was Xaybouathong District in Khammouane Province, Lao PDR. Khammouane Province is a medium-sized region located in central Lao PDR with a population of approximately 445,000. It is bordered by Vietnam to the south and Thailand to the west, and serves as a key transit point, next to Savannakhet in the south, with the third Friendship Bridge crossing the Mekong River to connect with Thailand. Xaybouathong District, which is characterized by small rural areas, is one of the 10 districts in Khammouane Province. About half the 100 km distance from the provincial capital, Thakhek District, to Xaybouathong District consists of unpaved roads. According to the local health office, the population of Xaybouathong District is approximately 32,000, with most being engaged in subsistence agriculture. The main ethnic groups include the Pouthai and Markon, with a minority of Lao people who migrated from other regions. The language spoken is exclusively Lao.

Due to the limited survey period, the selection of the research participants considered the feasibility of the survey based on the availability of local collaborators and access to villages. This led to eight of 65 villages being selected from Xaybouathong District (**Fig 1**). Considering that issues related to MHM are more pronounced among young women [14], the target population was women aged 15–24 years. Referring to information held by the Xaybouathong District Health Office, from approximately 421 registered individuals from the target villages, 80 were randomly selected to be surveyed.

### Questionnaire development and data collection

In the questionnaire survey, five questions were set regarding the sociodemographic and economic characteristics of the target women. For MHM, referring to previous studies conducted in Lao PDR [16, 18], three items on menstrual awareness, including secrecy, eight items on menstrual symptoms and responses, and three items on social support, totaling 14 questions, were prepared (**Table 1**). Regarding menstrual awareness, free-text responses about life problems caused by menstruation were included. Regarding social support, free-text responses about support needs were included. The questionnaire was initially created in English and then translated into Lao by local translators. To improve the accuracy of the questions, we conducted a pretest among residents outside of the survey population in Xaybouathong District to ensure that the questionnaire was accurately translated, understandable, and accessible.

Because many residents in the research area are involved in rice farming, the survey was conducted in March 2023 (the dry season) to avoid the busy farming period. On the day before the survey, information about the participants was communicated to collaborators such as

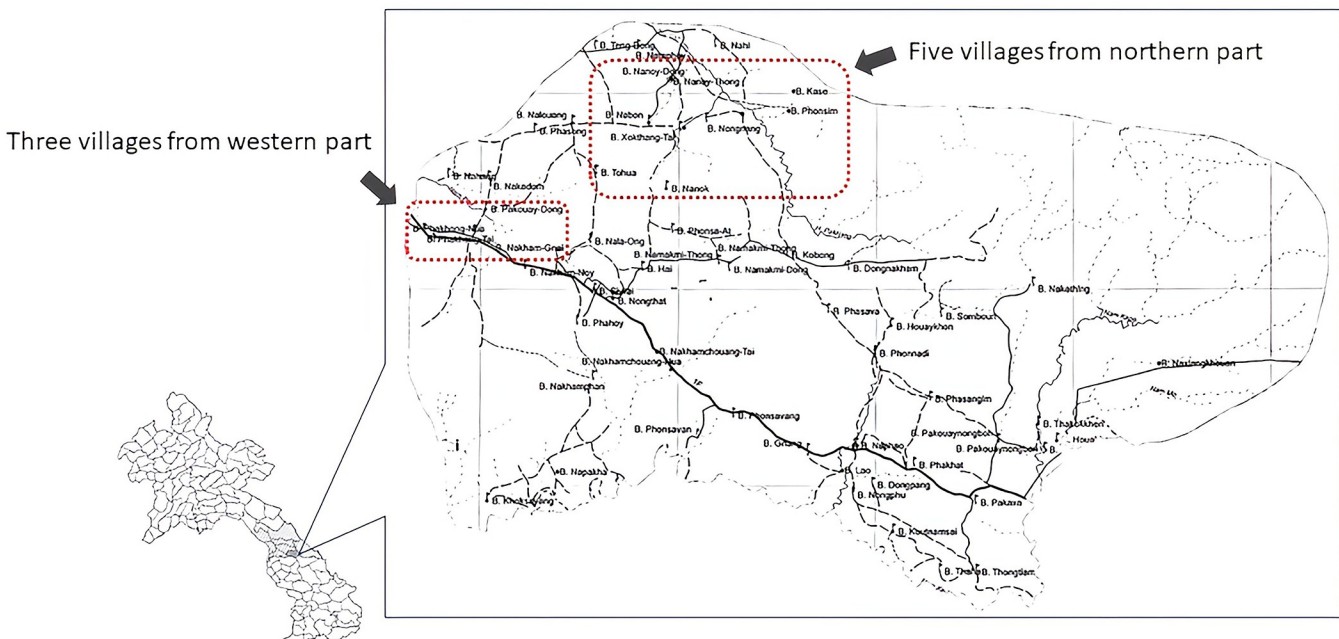

**Fig 1. Map of the target villages in Xaybouathong District, Khammouane Province.** It depicts the geographical characteristics of the study area within the target country, as well as the regions from which the sampling subjects for this study were obtained.

village leaders. On the survey day, village leaders helped to gather participants at community centers or schools. After explaining the purpose of the survey and emphasizing that participation was voluntary, only those willing to participate were asked to stay. Questionnaires were then distributed to the participants, and instructions were given on how to respond independently without consulting others. No time limit was given for answering the questions. The participants completed the questionnaire directly and received additional explanations individually only if they needed help understanding any questions. The questionnaires were collected after each participant finished and left the facility.

The implementation of the survey and the explanation to the participants were conducted in Lao by an interpreter (English-Lao) trained by the lead author (Kanayo Ono) beforehand. Additionally, throughout the entire survey period, staff members from the Xaybouathong District Health Office (health-care professionals) accompanied the team.

## Data analysis

We described the distribution of the five items on sociodemographic and economic characteristics and 12 quantitative questions about MHM obtained from the questionnaire responses through simple tabulation. Additionally, responses from two open-ended questions were classified by the lead author using the KJ method [19], where each group was named and finalized through a discussion among all authors, using the respondents' answers as labels. For the statistical analysis, we performed cross-tabulation with binary responses ("disagree" = 0, "agree" = 1) to the statement "I should not talk to others about menstruation" as the objective variable and responses related to MHM, such as symptoms, treatments, and social support, as explanatory variables. We assessed the relationships between explanatory variables to check for collinearity using chi-square tests, and we selected variables that showed a strong association, defined as having a p-value of 0.01 or less. Subsequently, we conducted bivariate logistic

**Table 1. Questions items on MHM perception and practices.**

| No | Theme | Questions |
|---|---|---|
| **Perception** | | |
| 1 | Secrecy | Do you agree that "I should not talk to others about menstruation"? [Not Agree / Agree] |
| 2 | Satisfaction | Are you satisfied with your current method of coping with your menstruation? [No / Yes] |
| 3 | Obstruction | How does menstruation obstruct your everyday activity? [Free-answered question] |
| **Symptoms and Treatments** | | |
| 4 | Menarche | At what age did you first experience menstruation? [less than 13 or more than 14] |
| 5 | Pain Level | What is the level of your pain caused by menstruation without any intervention? [1 to 4] |
| 6 | Pain Control | How do you usually control your pain caused by menstruation? [Choose option] *multiple answer |
| 7 | Cleanness | How do you usually maintain cleanness your pubic area during menstruation? [Choose option] *multiple answer |
| 8 | Blood Treatment | How do you usually treat your menstrual blood? [Choose option] *multiple answer |
| 9 | Private Place Availability (at Home) | Do you usually use a private place at HOME for changing sanitary materials? [No / Yes] |
| 10 | Private Place Availability (outside Home) | Are you usually available to change sanitary materials in a private place OUTSIDE the home? [No / Yes] |
| 11 | Waste Bins | Are you usually available in bins OUTSIDE the home to dispose of sanitary materials? [Yes / No] |
| **Social Supports** | | |
| 12 | Talk with | Who can you talk to about menstruation among the following? [Choose option] *multiple answer |
| 13 | Learn from | Who did you learn about managing menstruation from? [Choose option] *multiple answer |
| 14 | Support | What kind of support do you think would be helpful regarding menstruation? [Free-answered question] |

regression analyses to examine the association between the objective variable and the selected explanatory variables. Statistical analysis was conducted with 95% confidence interval (CI) and a significance level of 5%. Data from the questionnaire were inputted using Microsoft Excel, and R [20] was used for the statistical analysis.

## Ethics approval and informed consent

This study was approved by the Research Ethics and Safety Committee of Oita Prefectural College of Nursing (Approval No.: 22–63) and the Research Ethics Review Committee in Lao PDR (Approval No.: 2022.77). Informed consent was obtained from all participants after ensuring that they were informed about the study purpose, methodology, questionnaire items, and use and disposal of collected data. All participants were assured that their cooperation was voluntary and that they could refuse to participate without facing any inconvenience. It was

also explained that the research results would be published without disclosing the identity of individuals or specific regions. For participants under 18 years of age, both verbal and written explanations were provided to their guardians, and consent was obtained through written consent forms. If consent was not obtained, that individual was not included in the study.

## Results

### Characteristics of target women

From among the 80 targeted women, 70 (87.5%) provided valid responses. The average age of the respondents was 19.8 years, with 64.3% having completed secondary education or higher, followed by 31.4% with primary education, and 4.3% with no formal education (Table 2). Most of the participants were living with their parents (67.1%), followed by a spouse (37.1%), siblings (37.1%), and grandparents (11.4%). Regarding monthly disposable income, the most common range was 60,000–100,000 LAK (approximately 3.51–5.86 USD), reported by 62.9% of the respondents.

### MHM situation among the targeted women

Table 3 presents the results regarding the current status of MHM. The percentage of participants who agreed with the statement "I should not talk to others about menstruation" was 38.6%. Furthermore, regarding current satisfaction with MHM, 97.1% responded that they were "satisfied." Table 4 shows the results of the free-text responses regarding "daily life inhibited by menstruation" obtained through the KJ method. All 70 respondents provided some

**Table 2. Sociodemographic and economic characteristics of the targeted women.**

| Items | Respondents (N = 70) (%) |
|---|---:|
| Age* | 19.8 (±2.9) |
| ≤ 19 years old | 34 (48.6) |
| 20 years old ≤ | 36 (51.4) |
| Residential Area | |
| Northern Part | 44 (62.9) |
| Western Part | 26 (37.1) |
| Educational Attainments | |
| No Education | 3 (4.3) |
| Primary | 22 (31.4) |
| Secondary or more | 45 (64.3) |
| Family status living together[†] | |
| Live with parents | 47 (67.1) |
| Live with husband | 26 (37.1) |
| Live with grand parents | 8 (11.4) |
| Live with siblings | 26 (37.1) |
| Allowance (per month)[‡] | |
| ≤ 50,000 kip | 10 (14.3) |
| 50,001–100,000 kip | 44 (62.9) |
| 100,000 kip ≤ | 16 (22.9) |

* Means (Standard Deviation)

[†] Multiple response

[‡] 50,000₭ = 2.93$, 100,000₭ = 5.86$

**Table 3. Results of MHM perception and practices among the targeted women.**

| Items | Respondents (N = 70) (%) |
|---|---|
| **Perception** | |
| Secrecy | |
| Yes | 27 (38.6) |
| Satisfaction | |
| Yes | 68 (97.1) |
| **Symptoms and Treatments** | |
| Menarche | |
| $\leq$ 13 years old | 28 (40.0) |
| 14 years old $\leq$ | 42 (60.0) |
| Pain Level | |
| No pain | 16 (22.9) |
| Little | 21 (30.0) |
| Mild | 18 (25.7) |
| Severe | 15 (21.4) |
| Pain Control* | |
| • Painkiller | 20 (28.6) |
| • Herb | 3 (4.3) |
| • Warming | 19 (27.1) |
| • Do nothing | 26 (37.1) |
| Cleanness* | |
| • Cleaning by water with soap | 52 (74.3) |
| • Cleaning by water | 18 (25.7) |
| • Cleaning by paper | 10 (14.3) |
| Blood Treatment* | |
| • Disposal Napkin | 66 (94.3) |
| • Paper | 4 (5.7) |
| • Tampon | 1 (1.4) |
| • No treatment | 2 (2.9) |
| Private Place Availability (at Home) | |
| Yes | 43 (61.4) |
| Private Place Availability (outside Home) | |
| Yes | 49 (70.0) |
| Waste Place Availability | |
| Yes | 52 (74.3) |
| **Social Supports** | |
| Talk with* | |
| • Mother | 46 (65.7) |
| • Older Sister | 16 (22.9) |
| • Friends | 1 (1.4) |
| • Relatives | 1 (1.4) |
| • Teachers | 19 (27.1) |
| • Medical Professionals | 1 (1.4) |
| • Others | 1 (1.4) |
| Learn from* | |
| • Mother | 41 (58.6) |
| • Older Sister | 21 (30.0) |

(*Continued*)

**Table 3.** (Continued)

| Items | Respondents<br>(N = 70) (%) |
|---|---|
| • Friends | 25 (35.7) |
| • Relatives | 1 (1.4) |
| • Teachers | 1 (1.4) |
| • Medical Professionals | 0 (0.0) |
| • Never | 1 (1.4) |

\* Multiple response

information, and six categories were extracted. The most frequently extracted category was "able to live as usual," followed by "difficulties in working," "unable to move," "sleepiness," and "nausea." In the category "difficulties in working," many responses mentioned the inability to work hard, indicating that the content of hindrance to daily life due to menstruation was mainly associated with labor.

Regarding responses about menstrual symptoms and management, 60.0% of the participants reported experiencing their first menstruation at age 14 years or later, although all participants had experienced menarche. Concerning symptoms, 22.9% reported having no pain, 30.0% mild pain, 25.7% moderate pain, and 21.4% severe pain. The most common pain management method was doing nothing (37.1%), followed by taking painkillers orally (28.6%), applying heat to the abdomen (27.1%), and taking time off school/work (20.0%). Regarding menstrual hygiene, the majority of the respondents (74.3%) reported using both soap and water for cleaning, while some used water alone (25.7%) or wiped with paper only (14.3%). Some additional responses indicated beliefs such as not washing their hair during menstruation to prevent "unclean" menstrual blood from flowing and performing hygiene only with paper, including for the genital area. Concerning menstrual blood disposal, 94.3% of the respondents reported using disposable (nonreusable) pads. A small percentage of the respondents (2.9%) reported not using any menstrual products (or being unable to purchase them), adding that during menstruation, they try to lie down and avoid movement as much as possible.

Regarding the availability of facilities for MHM, 38.6% of the respondents stated that they lacked a space at home to ensure privacy for changing menstrual products. This percentage decreased outdoors, with 30.0% stating no available space. Additionally, 25.7% reported needing a designated place outdoors to dispose of menstrual products. When asked further about menstrual product disposal, some respondents stated that a disposal location mentioned burying used pads in the ground. Some women even reported walking to the forest for 2 hours to dispose of them, leading to work or school disruptions.

**Table 4. Summary of daily life activities restricted by menstruation using the KJ method.**

| Categories | Frequency |
|---|---|
| No problems | 31 |
| Unable to work as usual | 25 |
| Unable to move | 16 |
| Drowsiness | 2 |
| Feeling unwell | 2 |
| Others | 1 |

**Table 5. Summary of support needed for menstruation using the KJ method.**

| Categories | Frequency |
|---|---|
| No idea | 25 |
| Method for coping with abdominal pain | 23 |
| No need any support | 22 |
| Teaching about menstruation in School | 3 |
| Treatment of other symptoms | 1 |

In terms of social support for MHM, the individuals most commonly mentioned as someone the respondents could talk to about menstruation were mothers (65.7%), followed by teachers (27.1%), and older sisters (22.9%); friends and health-care professionals were mentioned less frequently. Mothers (58.6%), friends (35.7%), and older sisters (30.0%) were the most commonly cited individuals the respondents relied on as sources of information about menstruation, whereas teachers and health-care professionals were mentioned less often. Older relatives such as grandparents and male figures such as husbands or fathers were not mentioned as either sources of information or confidants.

## Need for menstruation support

All 70 respondents provided some information in the free-text responses regarding the question "What support is needed to manage daily life during menstruation?" As a result of categorizing the responses using the KJ method, five categories were identified (**Table 5**). The most frequent category was "Do not know," with responses from 25 individuals (35.7%), followed by "Dealing with abdominal pain," with responses from 23 individuals (32.9%). Other categories included "No particular support needed," "Providing menstrual information in educational institutions," and "Dealing with symptoms other than abdominal pain".

## Relationship between menstrual secrecy and MHM practices

The respondents agreeing and disagreeing with menstrual secrecy were divided into two groups of 27 (38.6%) and 43 individuals (61.4%), respectively. The results of the statistical analysis are presented in **Table 6**. Regarding the association with sociodemographic and economic characteristics, having more disposable monthly income was associated with a tendency to not keep menstrual secrecy (odds ratio: 0.15, 95% CI: 0.02–0.85). No significant associations were found between any of the MHM's Symptoms and Treatments and Social Supports.

## Discussion

### MHM practice among young women living in rural Lao PDR

The findings of the present study shed light on the reality of MHM among young women living in rural areas of Khammoane Province, Lao PDR.

Compared with other selected studies conducted in Lao PDR [16, 17], while demographic characteristics such as education level appeared to have similar distributions, a tendency for a higher amount of disposable income per month was observed. This could be attributed to the fact that this study included female students and adults up to age 24 years as participants. Additionally, some participants were not living with their parents, but rather, were already married and living with their husbands, which may have contributed to the variance in disposable income compared with studies targeting teenage students exclusively.

**Table 6. Results of a bivariate logistic regression analysis on the relationship between menstrual secrecy and MHM practices.**

| Items | Menstrual Secrecy | | OR | 95% CI | P-value |
|---|---|---|---|---|---|
| | Not Agree (N = 43) (%) | Agree (N = 27) (%) | | | |
| **Age** | | | | | |
| ≤ 19 years old | 17 (39.5) | 17 (63.0) | Ref | Ref | Ref |
| 20 years old ≤ | 26 (60.5) | 10 (37.0) | 0.38 | [0.14–1.02] | 0.059 |
| **Residential Area** | | | | | |
| Northern Part | 27 (62.8) | 17 (63.0) | Ref | Ref | Ref |
| Western Part | 16 (37.2) | 10 (37.0) | 0.99 | [0.36–2.68] | 0.988 |
| **Educational Attainments** | | | | | |
| No Education or Primary | 16 (37.2) | 9 (33.3) | Ref | Ref | Ref |
| Secondary or more | 27 (62.8) | 18 (66.7) | 1.19 | [0.44–3.34] | 0.742 |
| **Allowance (per month)** | | | | | |
| ≤ 50,000 kip | 4 (9.3) | 6 (22.2) | Ref | Ref | Ref |
| 50,001–100,000 kip | 26 (60.5) | 18 (66.7) | 0.46 | [0.15–1.85] | 0.279 |
| 100,000 kip ≤ | 13 (37.1) | 3 (11.1) | 0.15 | [0.02–0.85] | 0.040 |
| **Satisfaction** | | | | | |
| No | 1 (2.3) | 1 (3.7) | Ref | Ref | Ref |
| Yes | 42 (97.7) | 26 (96.3) | 0.62 | [0.02–16.10] | 0.738 |
| **Menarche** | | | | | |
| ≤ 13 years old | 14 (32.6) | 14 (51.9) | Ref | Ref | Ref |
| 14 years old ≤ | 29 (67.4) | 13 (48.1) | 0.45 | [0.16–1.20] | 0.112 |
| **Pain Level** | | | | | |
| No pain | 8 (18.6) | 8 (29.6) | Ref | Ref | Ref |
| Little | 15 (34.9) | 6 (22.2) | 0.40 | [0.98–1.54] | 0.188 |
| Mild | 9 (20.9) | 9 (33.3) | 1.00 | [0.26–3.90] | 1.000 |
| Severe | 11 (25.6) | 4 (14.8) | 0.36 | [0.07–1.58] | 0.188 |
| **Pain Control** | | | | | |
| • Painker | | | | | |
| No | 31 (72.1) | 19 (70.4) | Ref | Ref | Ref |
| Yes | 12 (27.9) | 8 (29.6) | 1.09 | [0.37–3.13] | 0.877 |
| • School absence / Day-off work | | | | | |
| No | 35 (81.4) | 22 (81.5) | Ref | Ref | Ref |
| Yes | 8 (18.6) | 5 (18.5) | 0.99 | [0.27–3.38] | 0.993 |
| **Cleanness** | | | | | |
| • Cleaning by water with soap | | | | | |
| No | 9 (20.9) | 9 (33.3) | Ref | Ref | Ref |
| Yes | 34 (79.1) | 18 (66.7) | 0.53 | [0.18–1.58] | 0.251 |
| **Blood Treatment** | | | | | |
| • Disposal Napkin | | | | | |
| No | 3 (7.0) | 1 (3.7) | Ref | Ref | Ref |
| Yes | 40 (93.0) | 26 (96.3) | 1.95 | [0.24–40.59] | 0.572 |
| **Private Place Availability (at Home)** | | | | | |
| No | 18 (41.9) | 9 (33.3) | Ref | Ref | Ref |
| Yes | 25 (58.1) | 18 (66.7) | 1.44 | [0.53–4.04] | 0.476 |
| **Private Place Availability (outside Home)** | | | | | |
| No | 16 (37.2) | 5 (18.5) | Ref | Ref | Ref |
| Yes | 27 (62.8) | 22 (81.5) | 2.61 | [0.87–7.00] | 0.103 |

*(Continued)*

**Table 6.** (Continued)

| Items | Menstrual Secrecy | | OR | 95% CI | P-value |
|---|---|---|---|---|---|
| | Not Agree (N = 43) (%) | Agree (N = 27) (%) | | | |
| **Talk with** | | | | | |
| • Mother | | | | | |
| No | 14 (32.6) | 10 (37.0) | Ref | Ref | Ref |
| Yes | 29 (67.4) | 17 (63.0) | 0.82 | [0.30–2.28] | 0.701 |
| • Teachers | | | | | |
| No | 32 (74.4) | 19 (70.4) | Ref | Ref | Ref |
| Yes | 11 (25.6) | 8 (29.6) | 1.22 | [0.41–3.58] | 0.711 |

Ref; Reference OR; Odds Ratio CI; Confidence Interval

In terms of MHM-related aspects, nearly 80% of the participants reported experiencing menstrual pain, including mild discomfort. Standard methods for pain management included applying heat to the abdomen and taking pain medication. The use of herbal remedies was reported in a previous study [3], but in the present study, this was only reported by 4.3% of the participants. Approximately 20% of the participants reported taking time off from school or work to address their symptoms. However, insights from the open-ended responses revealed that even those who did not take time off struggled to maintain their usual performance during menstruation. The association between menstrual pain and quality of life suggests a high need for medical care [21].

Over 90% of the participants reported using commercially available disposable pads for menstrual blood disposal. This aligns with usage rates in the urban areas of Lao PDR, as per a national survey [14]. However, a small number of the respondents reported using paper to manage menstrual blood or being unable to afford menstrual products. Nearly 40% of the participants reported that access to private spaces for changing pads during menstruation was difficult at home. This is likely due to the nature of housing in Lao PDR, where adequately partitioned rooms may be limited.

Conversely, the percentage of respondents finding it challenging to secure private spaces outdoors was about 30%. This could be because Lao PDR is known to dispose of waste in bushes or forests, which offers more privacy outdoors. However, the fact that over 30% of the respondents reported being unable to fulfill the MHM definition of "using clean menstrual products and changing them as needed in a private space" suggests the need for improvement.

Over 70% of participants reported having access to menstrual product disposal. However, this does not necessarily indicate access to *APPROPRIATE* disposal facilities; instead, it reflects situations where participants mentioned disposing of menstrual products by "burying them in the forest." Lessons from initiatives distributing pads in India underscore the necessity of support for disposal methods [22]. Disposable pads consist of materials like the surface layer, which is in contact with the skin, an absorbent core made of cotton pulp and superabsorbent polymers, a leak-proof barrier, and adhesive strips, none of which can safely return to the soil. Additionally, research from Nigeria suggests that menstrual waste holds cultural significance and needs to be disposed of carefully to avoid being used in rituals or magic [13]. Understanding the significance of burying pads in the forest in the context of the study area requires further investigation to elucidate the actual practices.

While proper MHM, including the disposal of menstrual blood and maintaining cleanliness, is crucial for preventing gynecological diseases [23, 24], only about 70% of the

participants reported washing their bodies and lower abdomens with soap and water during menstruation. Moreover, all of the participants needed to practice adequate hygiene measures. Although this study did not thoroughly examine the reasons for restricted access to water and soap, some participants mentioned believing that washing their hair during menstruation would lead to the impurity of menstrual blood not flowing out, reflecting certain social and cultural beliefs and practices. Further investigation is needed to determine whether these sociocultural beliefs and behaviors are associated with physical health risks.

Regarding social support, most participants reported consulting their mothers regarding menstruation, aligning with similar trends observed in previous studies. Additionally, approximately 30% of the participants mentioned consulting their schoolteachers, likely because they were attending school around the time of menarche. Conversely, consulting health-care professionals or male relatives was less common, reflecting a tendency to rely on female peers for menstruation-related issues, consistent with findings from previous research [8]. Regarding sources of information about menstruation, mothers, friends, and older sisters were the most commonly cited. Although mentioned as the second most consulted party, teachers were not selected as a source of information. Given that misinformation about menstruation can lead to inappropriate MHM practices [25], the absence of teachers and health-care professionals as information sources presents a challenge in terms of disseminating accurate information.

A study conducted in Saudi Arabia reported that issues related to menstruation experienced by women, such as menstrual pain or premenstrual syndrome, are associated with perceived stress [26]. However, among the participants in the present study, although it became evident that they were experiencing problems related to menstruation or could not achieve proper MHM, their overall satisfaction with current menstrual management was high (97.1%). This discrepancy might be attributed to a need for correct understanding or knowledge about menstruation and the normalization of the current environment, which makes it difficult for individuals to perceive dissatisfaction or challenges with the current situation. Therefore, there is a need to disseminate knowledge and consider improving comprehensive MHM environments [27].

## Menstrual secrecy and MHM practices

No statistically significant associations were observed regarding the relationship between menstrual secrecy and MHM practices. In other words, individuals who believe they should not discuss menstruation with others and those who do not share this belief exhibit similar levels of MHM. However, a correlation was observed with an economic factor: the amount of monthly disposable income available for discretionary spending. A trend was noticed where individuals with more discretionary income were less likely to agree with the statement, "I should not talk to others about menstruation." Previous studies targeting adolescents have suggested that higher allowances are associated with better MHM practices [11, 28], indicating that individuals with greater financial freedom can afford the expenses associated with MHM. Consequently, feelings of shame or the need for secrecy regarding menstruation are likely lower among individuals with more financial autonomy.

## Limitations

One limitation of this study is the restricted survey period, which necessitated limiting the sample size and study area. Therefore, only the bivariate analysis was applied, and the effects between explanatory variables have not been thoroughly examined. Additionally, the results cannot be taken to represent the entire Xaybouathong District. Due to the scarcity of previous research on young women in Lao PDR and the broad focus of this study, it was challenging to

investigate each component of MHM. Consequently, detailed inquiries into various aspects of MHM could not be conducted. Moving forward, it will be imperative to expand and sustain research efforts to address the new challenges identified in this study.

## Conclusions

While previous research on MHM has reported sociocultural influences, no significant associations between the cultural value of menstruation secrecy and the implementation of MHM were found in the present study. In the limited environment of rural Lao PDR, it is evident that groups both with and without traditional values have achieved similar levels of MHM. However, both groups also experience limitations in their daily activities because of menstrual symptoms, struggle to maintain privacy during menstrual product changes, and do not always practice proper hygiene. Additionally, subjective satisfaction with current MHM is generally high. Therefore, in resource-constrained areas, it is beneficial to not only consider traditional values, but also understand how individuals are specifically perceiving, managing, and coping with the current state of MHM.

## Supporting information

**S1 Dataset.**
(XLSX)

## Acknowledgments

We would like to thank Dr. Somboune and Dr. Khampanavanh from the Khammouane Provincial Health Office for their approval of and cooperation with the local investigation. We also sincerely thank Mr. Niphop and Ms. Syntala from the Xaybouathong District Health Office for their coordination during the local survey. Our deepest thanks go to the chiefs of each target village for their assistance. We are also grateful to the research participants from Xaybouathong District and their parents for participating in this study. Lastly, we extend our heartfelt appreciation to Oita University of Nursing and Health Sciences and the International Society for the Advancement of Physical Healthcare (ISAPH) for granting permission for overseas research and providing us with this invaluable opportunity.

## Author Contributions

**Conceptualization:** Kanayo Ono, Noriko Kuwano, Kana Maruyama.

**Data curation:** Kanayo Ono, Hisao Ando.

**Formal analysis:** Yu Sato.

**Investigation:** Kanayo Ono, Hisao Ando.

**Methodology:** Kanayo Ono, Noriko Kuwano, Hisao Ando, Kana Maruyama.

**Supervision:** Yu Sato.

**Writing – original draft:** Kanayo Ono, Yu Sato.

**Writing – review & editing:** Kanayo Ono, Yu Sato, Noriko Kuwano, Hisao Ando, Kana Maruyama.

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
