## [Decision Letter · Decision Letter 0]

21 Aug 2024

PONE-D-24-15653Menstrual hygiene management and menstrual secrecy among young women in rural Lao PDR: A cross-sectional studyPLOS ONE

Dear Dr. SATO,

Thank you for submitting your manuscript to PLOS ONE. After careful consideration, we feel that it has merit but does not fully meet PLOS ONE’s publication criteria as it currently stands. Therefore, we invite you to submit a revised version of the manuscript that addresses the points raised during the review process.

We look forward to receiving your revised manuscript.

Kind regards,

Mubarick Nungbaso Asumah, MPhil, Bsc

Academic Editor

PLOS ONE

Journal Requirements:

2. In the online submission form, you indicated that "The data supporting the findings of this study are available upon reasonable request to the corresponding author, YS. However, please note that the data cannot be publicly disclosed due to the inclusion of personal information that could compromise the privacy of the research participants."

**Additional Editor Comments:**

Kindly respond to the reviewers comments and also ensured to incorporate the following papers in your. There are some missing link in your introduction, I believe this would help you refine the introduction well enough:

https://pubmed.ncbi.nlm.nih.gov/35519170/

https://bmjopen.bmj.com/content/12/4/e056526

https://pubmed.ncbi.nlm.nih.gov/35978966/

Reviewers' comments:

Reviewer's Responses to Questions

**Comments to the Author**

1. Is the manuscript technically sound, and do the data support the conclusions?

Reviewer #1: Yes

Reviewer #2: Yes

2. Has the statistical analysis been performed appropriately and rigorously? 

Reviewer #1: No

Reviewer #2: Yes

3. Have the authors made all data underlying the findings in their manuscript fully available?

Reviewer #1: Yes

Reviewer #2: Yes

4. Is the manuscript presented in an intelligible fashion and written in standard English?

Reviewer #1: Yes

Reviewer #2: Yes

5. Review Comments to the Author

Reviewer #1: COMMENTS FOR AUTHORS

1. There are minor changes in the work which require the authors to make the necessary changes. See the work and effect the needed changes.

2. The authors should also check the analysis of bivariate logistic regression. There are some technical data analytical challenges which should be looked at.

3. Again, the authors could have also conducted a chi-square analysis to examine the association among the studied variables before performing the logistic analysis

Reviewer #2: Detail my comments:

1. The title and the abstract accurately and concisely summarises

2. The content of the study provides new insight into which aspects need to be considered for promoting MHM in addition to information for future initiatives

3. Keywords: please add them from a term mesh, put them, and locate them alphabetically.

4. The study context is well described and the research method is clear

5. The method and data of analysis are clearly described

6. The research relates the findings to previous work as well as their implications for practice, however, the cultural value of menstruation secrecy and the implementation of MHM are important to discuss or elaborate more on the temporal trend, in deep interview for the next study

7. The limitations of this study are appropriate

6. PLOS authors have the option to publish the peer review history of their article (what does this mean?). If published, this will include your full peer review and any attached files.

Reviewer #1: No

Reviewer #2: **Yes: **Yati Afiyanti

---

## [Author Response · Author response to Decision Letter 0]

26 Aug 2024

We are sending the file "Response to Reviewers." Here is our quick response; please see it for the details.

Journal Requirements:

  I have carefully reviewed the information in the template and made the necessary revisions.

2. This policy applies to all data except where public deposition would breach compliance with the protocol approved by your research ethics board. 

 we have confirmed that our final dataset cannot be personally identified of the participants. Therefore, we agree to make the data available as an open source (as supplementary information).

3. Please review your reference list to ensure that it is complete and correct.

 In accordance with the template, we have revised the format of the reference list and ensured that all entries are complete and accurate.

Additional Editor Comments:

Kindly respond to the reviewers comments and also ensured to incorporate the following papers in yours.

 One of the papers was already cited in the background section (Line 63). We have added the other two papers you suggested to ensure a comprehensive introduction (Lines 46-48 & Lines 66-71).

Reviewers' comments:

Reviewer #1: COMMENTS FOR AUTHORS

1. There are minor changes in the work which require the authors to make the necessary changes. See the work and effect the needed changes.

2. The authors should also check the analysis of bivariate logistic regression. There are some technical data analytical challenges which should be looked at.

3. Again, the authors could have also conducted a chi-square analysis to examine the association among the studied variables before performing the logistic analysis

 In line with your suggestions, we performed chi-square tests to examine the relationships between variables before conducting logistic regression analysis, ensuring that only the necessary variables were included in the analysis (Lines 171-175). Consequently, we have also revised Table 6 (Line 268).

 Considering appropriate statistical analyses, multivariate analysis using the selected variables would be more suitable. However, as noted in the study limitations, there is a risk of overfitting. Therefore, we did not conduct multivariate analysis in this study. This point has been clarified in the study limitations (Lines 365-369).

Reviewer #2: Detail my comments:

1. The title and the abstract accurately and concisely summarises

2. The content of the study provides new insight into which aspects need to be considered for promoting MHM in addition to information for future initiatives

3. Keywords: please add them from a term mesh, put them, and locate them alphabetically.

4. The study context is well described and the research method is clear

5. The method and data of analysis are clearly described

6. The research relates the findings to previous work as well as their implications for practice, however, the cultural value of menstruation secrecy and the implementation of MHM are important to discuss or elaborate more on the temporal trend, in deep interview for the next study

7. The limitations of this study are appropriate

 Following your suggestions, we have added keywords from MeSH terms and arranged them in alphabetical order. As for the socio-cultural significance of menstrual secrecy, we will continue to conduct field research on this topic as part of our future research agenda.

---

## [Editor Report · Decision Letter 1]

12 Sep 2024

Menstrual hygiene management and menstrual secrecy among young women in rural Lao PDR: A cross-sectional study

PONE-D-24-15653R1

Dear Dr. SATO,

We’re pleased to inform you that your manuscript has been judged scientifically suitable for publication and will be formally accepted for publication once it meets all outstanding technical requirements.

Kind regards,

Mubarick Nungbaso Asumah, MPhil, Bsc

Academic Editor

PLOS ONE

Additional Editor Comments (optional):

While we were unable to have the original reviewers evaluate the authors' responses, I have carefully reviewed them and find the responses to be satisfactory and comprehensive.
---

## [Editor Report · Acceptance letter]

16 Sep 2024

PONE-D-24-15653R1 

PLOS ONE

Dear Dr. Sato, 

I'm pleased to inform you that your manuscript has been deemed suitable for publication in PLOS ONE. Congratulations! Your manuscript is now being handed over to our production team.

Kind regards, 

on behalf of

Dr. Mubarick Nungbaso Asumah 

Academic Editor

PLOS ONE